# Phenotype-based probabilistic analysis of heterogeneous responses to cancer drugs and their combination efficacy

Natacha Comandante-Lou[1], Mehwish Khaliq[1,2], Divya Venkat[3], Mohan Manikkam[1], Mohammad Fallahi-Sichani[1,2,4]*

**1** Department of Biomedical Engineering, University of Michigan Medical School, Ann Arbor, Michigan, United States of America, **2** Program in Cancer Biology, University of Michigan Medical School, Ann Arbor, Michigan, United States of America, **3** Department of Biochemistry, University of Michigan Medical School, Ann Arbor, Michigan, United States of America, **4** Department of Dermatology, University of Michigan, Ann Arbor, Michigan, United States of America

* fallahi@umich.edu

**Data Availability Statement:** All relevant data are within the manuscript and its Supporting Information files.

## Abstract

Cell-to-cell variability generates subpopulations of drug-tolerant cells that diminish the efficacy of cancer drugs. Efficacious combination therapies are thus needed to block drug-tolerant cells via minimizing the impact of heterogeneity. Probabilistic models such as Bliss independence have been developed to evaluate drug interactions and their combination efficacy based on probabilities of specific actions mediated by drugs individually and in combination. In practice, however, these models are often applied to conventional dose-response curves in which a normalized parameter with a value between zero and one, generally referred to as fraction of cells affected ($f_a$), is used to evaluate the efficacy of drugs and their combined interactions. We use basic probability theory, computer simulations, time-lapse live cell microscopy, and single-cell analysis to show that $f_a$ metrics may bias our assessment of drug efficacy and combination effectiveness. This bias may be corrected when dynamic probabilities of drug-induced phenotypic events, i.e. induction of cell death and inhibition of division, at a single-cell level are used as metrics to assess drug efficacy. Probabilistic phenotype metrics offer the following three benefits. First, in contrast to the commonly used $f_a$ metrics, they directly represent probabilities of drug action in a cell population. Therefore, they deconvolve differential degrees of drug effect on tumor cell killing versus inhibition of cell division, which may not be correlated for many drugs. Second, they increase the sensitivity of short-term drug response assays to cell-to-cell heterogeneities and the presence of drug-tolerant subpopulations. Third, their probabilistic nature allows them to be used directly in unbiased evaluation of synergistic efficacy in drug combinations using probabilistic models such as Bliss independence. Altogether, we envision that probabilistic analysis of single-cell phenotypes complements currently available assays via improving our understanding of heterogeneity in drug response, thereby facilitating the discovery of more efficacious combination therapies to block drug-tolerant cells.

**Funding:** This work was supported by awards from the Elsa Pardee Foundation and V Foundation for Cancer Research V2017-011, Department of Defense PRCRP Career Development Award W81XWH1810427, NIH grants R00-CA194163 and R35-GM133404 (to MFS), P30-CA046592 (University of Michigan Rogel Cancer Center), Rackham International Student Fellowship (to NCL), and NCI Training Grant award T32-CA009676 (to MK). The funders had no role in study design, data collection and analysis, decision to publish, or preparation of the manuscript.

**Competing interests:** The authors have declared that no competing interests exist.

## Author summary

Resistance to therapy due to tumor cell heterogeneity poses a major challenge to the use of cancer drugs. Cell-to-cell variability generates subpopulations of drug-tolerant cells that diminish therapeutic efficacy, even in populations of cells that are scored as highly sensitive based on drug potency. Overcoming such heterogeneity and blocking subpopulations of drug-tolerant cells motivate efforts toward identifying efficacious combination therapies. The success of these efforts depends on our ability to distinguish how heterogeneous populations of cells respond to individual drugs, and how these responses are influenced by combined drug interactions. In this paper, we propose mathematical and experimental frameworks to evaluate time-dependent drug interactions based on probabilistic metrics that quantify drug-induced tumor cell killing or inhibition of division at a single-cell level. These metrics can reveal heterogeneous drug responses and their changes with time and drug combinations. Thus, they have important implications for designing efficacious combination therapies, especially those designed to block or overcome drug-tolerant subpopulations of cancer cells.

## Introduction

In pre-clinical studies, potentially effective drug combinations are usually identified based on evidence of synergy [1–4]. In the case of cancer drugs, synergistic interactions are typically assessed on the basis of bulk cell population measurements, such as relative viability (normalized cell count) and net growth rate inhibition, and their variations with drug dose and combination [5–9]. The benefit of drug combination is then evaluated based on whether using two drugs together improves the potency (via minimizing the dose) or efficacy of treatment (via enhancing the effect) as compared with using either of the drugs alone [10–16]. Such benefit with respect to efficacy and potency, however, may be decoupled [10], as each metric encodes distinct information about cellular response to a drug [17]. Variations in potency are often explained by differences in target engagement (e.g. physicochemistry of drug-target interaction), concentration of drug available to cells (e.g. drug uptake and efflux), or existence of pathway redundancy (e.g. presence of a secondary oncogenic driver), among others [18]. Thus, a more potent drug combination enables engaging the target and achieving the desired effect in a cell population by using lower doses of treatment [19,20]. Efficacy, on the other hand, refers to the maximum response achievable using tolerable doses of a drug. A more efficacious drug or drug combination engages a larger proportion of cells [21,22]. Previous systematic studies have revealed that variation in cancer drug efficacy is associated with the extent of cell-to-cell variability in drug response [17,23], although such heterogeneity is not directly scored in most pre-clinical drug response assays.

Cell-to-cell variability may generate subpopulations of drug-tolerant cells that diminish cancer drug efficacy [24–30]. Heterogeneity is observed following the emergence of adaptive resistance or selection of resistant subclones even in populations of cells that are scored as highly responsive based on drug potency (e.g. $EC_{50}$ measurements) in routine 3 to 5-day assays. In such cases, while more than half (often as many as 90–99%) of cells may respond to treatment (depending on time and dose), the remaining cells give rise to a drug-insensitive subpopulation of survivors that may stay quiescent or divide slowly in the presence of drug [31]. Although not obvious from the most commonly used potency measurements, the emergence of such survivors limits therapeutic efficacy, leading to residual cells from which drug-resistant clones may eventually arise and drive disease progression [32–35].

Overcoming such heterogeneity in drug response and eradicating subpopulations of drug-tolerant cells provide a strong motivation for identifying more efficacious combination therapies [36]. A key step toward this goal is the ability to distinguish how heterogeneous populations of cells respond to individual drugs in short-term assays, and how these responses are influenced by combined drug interactions. However, the standard way in which drugs or their combinations are screened using normalized population assays obscures single-cell and subpopulation effects that likely play a major role in diminishing the therapeutic efficacy [21,37].

Focusing on efficacy, the benefit of drug combination in a heterogeneous population of cells may arise either from its cooperative inhibitory effect on target cells [22], or simply from the increased probability of cells being sensitive to any of the constituent drugs [38]. In both cases, the overall phenotypic consequences of drug interactions may be assessed in cell culture experiments based on null models of non-interaction [3]; synergistic efficacy is typically concluded when the observed combinatorial effect exceeds the expected effect from a given null model. The most commonly used model, Bliss independence, evaluates interactions based on the probability theory for statistically independent drug actions [16]. In cancer treatment, two basic phenotypic events affected by drug action are cell death and division. The effect of a drug on an individual cell changes the probability of its survival or division within a given time interval. However, current application of the Bliss independence typically uses fraction of cells affected ($f_a$), a number between zero and one defined based on relative viability or net growth rate inhibition normalized to an untreated control at a fixed timepoint, as drug effect [3,7]. We argue that this commonly used approach leads to a bias in the estimation of both drug efficacy and combination effectiveness in heterogeneous cell populations, especially when the ultimate goal is to block or eradicate small subpopulations of drug-tolerant cells. This is because $f_a$ quantities are not equal to the time-dependent probabilities at which cell death or inhibition of cell division are induced by a drug.

In this paper, we discuss evaluating time-dependent drug responses based on probabilistic metrics that quantify drug-induced tumor cell killing and inhibition of division at a single-cell level. Using these phenotype metrics, we re-evaluate criteria for statistical independence of drug interactions based on probability theory. Experimentally, phenotype metrics are measured using time-lapse live cell microscopy via monitoring cells engineered to express fluorescent reporters for nucleus identification (to distinguish live versus dead cells) and cell cycle progression (to score division events). As a proof of concept, we evaluate the performance of the metrics in two BRAF-mutant melanoma cell lines exposed to a range of targeted drugs which have been tested or proposed to be studied in combination with standard of care BRAF and MEK kinase inhibitors. Dynamic measurements of the phenotype metrics reveal distinctive responses of melanoma cells to drug combinations that may not be distinguishable when assessed based on conventional assays. This is because these metrics deconvolve differential degrees of drug effect on tumor cell killing versus inhibition of division, which are not necessarily correlated across various drug treatments and their combinations. Furthermore, these metrics increase the sensitivity of short-term drug response assays to cell-to-cell heterogeneities and thus the presence or emergence of drug-tolerant subpopulations, which are typically overlooked in conventional drug response assays.

## Results

### Probabilistic description of drug-induced phenotypic events

We model the arrival of phenotypic events, including cell division and death, in a given cell population as independent non-stationary *Poisson* processes with time-varying rate constants ($k_{event}$). These rate constants are linked to the actual probabilities ($P_{event}$) with which such

events occur in individual cells within a series of short time intervals ($dt$):

$$P_{event} = 1 - e^{-k_{event}dt} \approx k_{event}dt \qquad (1)$$

At a population level, the occurrence of these phenotypic events can be described by *Poisson* processes of which the time-dependent rates of occurrence are directly related to the probabilities of events at a single-cell level (Fig 1A). Therefore, the distribution of death and division

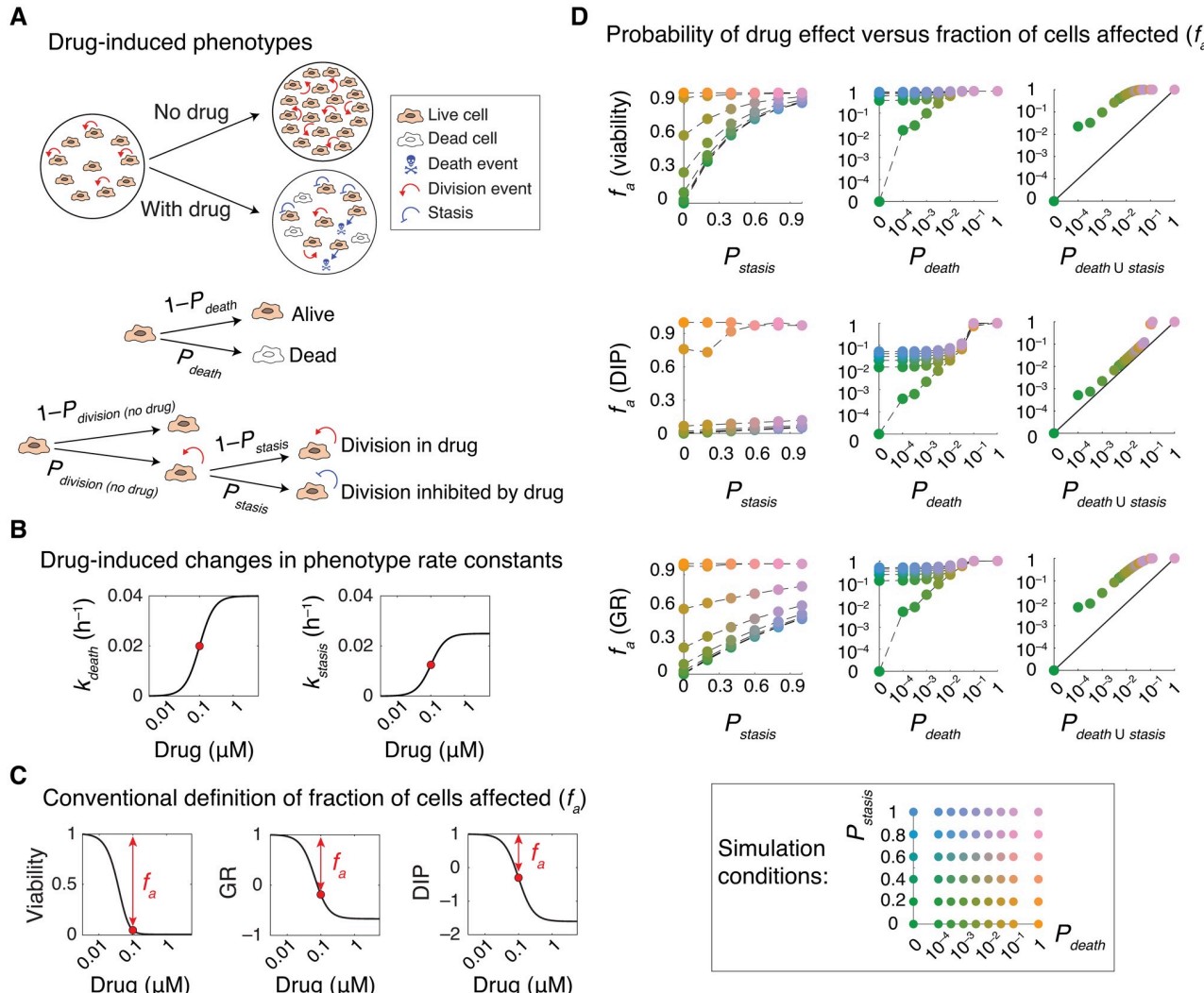

**Fig 1. Probabilistic description of drug-induced phenotypic events. (A)** Schematic representation of phenotypic effects of drug action in a cell population. Drug effect is described as probabilistic events, involving induction of cell death and inhibition of cell division, at a single-cell level. Cytotoxic effect of a drug on a given cell is described by the probability with which it induces cell death per unit of time ($P_{death}$). The cytostatic effect of drug on a given cell is described by a conditional probability ($P_{stasis}$) with which it prevents the cell from dividing given that the same cell would have divided in the absence of drug with a probability of $P_{division\ (no\ drug)}$. **(B)** Dose-dependent changes in phenotype rate constants ($k_{death}$ and $k_{stasis}$) in simulation of drug effect in a population of cells. **(C)** Model outputs showing variations in the fraction of cells affected ($f_a$) at t = 96 h corresponding to phenotype rate constant values shown in (B). $f_a$ may be calculated in three different ways based on bulk response metrics such as relative viability and net growth rate inhibition (GR and DIP) following normalization to an untreated control. **(D)** Simulation results comparing $f_a$ quantities at t = 96 h with probabilistic measures of drug action, $P_{death}$ quantified per unit of time (h), conditional probability $P_{stasis}$, and the overall probability with which a drug induces cell death or inhibits cell division ($P_{death\ \cup\ stasis}$) across a variety of conditions, representing drugs with different levels of cytotoxic and cytostatic effect. Each data-point represents the mean of 30 stochastic simulations. Cells grow from an initial number of $N_{live}$ (t = 0) = 1000 and at a rate of $k_{division\ (no\ drug)}$ = 0.025 h$^{-1}$.

events observed for a population of $N$ cells during a time period of $\Delta t$ may be approximated using the following equation:

$$Pr\{N_{event}(t \rightarrow t + \Delta t) = x\} = \frac{(k_{event}(t)N\Delta t)^x}{x!} e^{-(k_{event}(t)N)\Delta t} \qquad (2)$$

where $N_{event}(t \rightarrow t + \Delta t)$ is the number of phenotypic events (death or division) occurring during the time interval between $t$ and $t + \Delta t$.

Assuming negligible cell death in the absence of any treatment, the model describes the cytotoxic effect of a drug on a given cell by the probability with which it induces cell death per unit of time ($P_{death} = k_{death}dt$). The cytostatic effect of drug on a given cell is defined by a conditional probability ($P_{stasis}$) with which it prevents the cell from dividing given that the same cell would have divided in the absence of drug with a probability of $P_{division\ (no\ drug)} = k_{division\ (no\ drug)}dt$. The relationships between the conditional probability $P_{stasis}$ and the probability of cell division in the presence of drug ($P_{division\ (with\ drug)} = k_{division\ (with\ drug)}dt$) and their associated rate constants are as follows (see Materials and methods for the derivation details):

$$P_{stasis} = 1 - \frac{P_{division(with\ drug)}}{P_{division(no\ drug)}} = 1 - \frac{k_{division(with\ drug)}}{k_{division(no\ drug)}} \qquad (3)$$

$$k_{stasis} = P_{stasis}k_{division(no\ drug)} \qquad (4)$$

The model provides a framework to simulate how dose-dependent responses in populations of cells vary with $P_{death}$ per unit of time (h) and $P_{stasis}$ by using input parameters ($k_{death}$ and $k_{stasis}$) that represent drugs with a wide range of cytotoxic and cytostatic effects. For each condition, the fraction of cells affected ($f_a$), defined based on changes in relative viability or net growth rate inhibition (using recently developed drug-induced proliferation (DIP) and growth rate (GR) inhibition metrics [5,6]) normalized to an untreated control, are also derived as model outputs (Fig 1B and 1C). We compared $f_a$ quantities with probabilistic measures of drug action ($P_{death}$ and $P_{stasis}$) across a number of drug response simulations. Except for extreme cases such as when $P_{death} = P_{stasis} = 0$ (i.e. there is no drug) or when $P_{death} = 1$ (i.e. all cells dying within the first time interval), $f_a$ quantities differed substantially from the probability with which drugs induced cell death or from the probability with which they inhibited cell division (Fig 1D). $f_a$ gives a closer estimate of the overall probability with which a drug induces either cell death or inhibition of cell division ($P_{death\ \cup\ stasis}$), i.e. the probability of a cell being affected (Fig 1D). However, it still fails to accurately represent the probabilistic nature of drug action in cells. Together, simulation results suggest that using $f_a$ as a metric for probabilistic analysis of drug response or drug combination efficacy (such as in Bliss independence) might lead to unreliable conclusions. Instead, we propose to use direct measures of probabilistic phenotype metrics ($P_{death}$ and $P_{stasis}$ or $k_{death}$ and $k_{stasis}$) for such analyses.

## Probabilistic rate constants capture time-dependent heterogeneities in phenotypic responses

Probabilistic rate constants are estimated based on the frequencies of occurrence of individual phenotypic events. These metrics are expected to exhibit high sensitivity to the presence of cell-to-cell heterogeneities that cause the selection of small subpopulations of drug-tolerant cells. To test this hypothesis, we simulated drug treatment scenarios where the initial cell population consisted of heterogeneous subpopulations, in which a small fraction ($\omega \leq 5\%$) of cells were substantially less sensitive (by up to $r = 16$-fold) to treatment relative to the majority of the cell population (Fig 2A). We then defined and calculated "resistance enrichment ratio" for

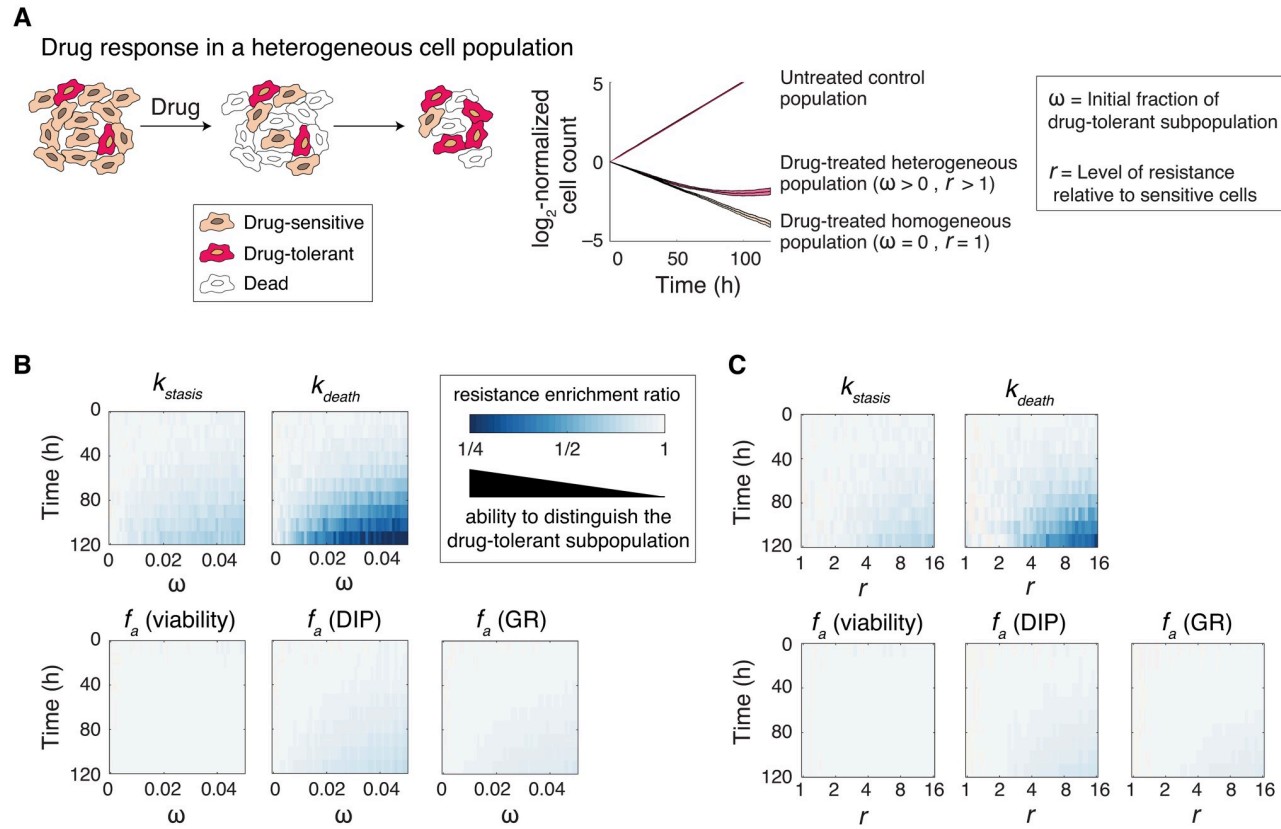

**Fig 2. Probabilistic rate constants capture time-dependent heterogeneities in phenotypic responses.** **(A)** Schematic representation of drug response in a heterogeneous cell population. Prior to drug-treatment, cells consist of a dominantly drug-sensitive population plus a small fraction ($\omega \leq 5\%$) of drug-tolerant subpopulation which is $r$ times more drug-resistant than the majority of cells. Upon drug treatment, the drug-tolerant subpopulation is gradually enriched over time. Resistance enrichment ratio for each of the $f_a$ metrics (described based on viability, GR and DIP) or for phenotype rate constants ($k_{death}$ and $k_{stasis}$) is calculated by normalizing each metric measured for a heterogeneous population to that in a homogeneous population (i.e. $\omega = 0$ or $r = 1$) at different times of treatment. Smaller resistance enrichment ratios represent greater sensitivity to the presence of heterogeneous drug-tolerant cells. **(B)** Simulation results showing changes in resistant enrichment ratio for each of the $f_a$ metrics or for phenotype rate constants ($k_{death}$ and $k_{stasis}$) as a function of time and $\omega$ (at a fixed value of $r = 16$). **(C)** Simulation results showing changes in resistant enrichment ratio for each of the $f_a$ metrics or for phenotype rate constants ($k_{death}$ and $k_{stasis}$) as a function of time and $r$ (at a fixed value of $\omega = 0.03$). Data shown are mean values from 50 simulations. In all simulations, we assumed fixed inherent growth rates for the sensitive and resistant populations: $k^{S}_{division\ (no\ drug)} = 0.035$ h$^{-1}$ and $k^{R}_{division\ (no\ drug)} = 0.02$ h$^{-1}$. All key parameters are described in Materials and Methods.

each of the $f_a$ metrics (described based on viability, GR and DIP) or for phenotype rate constants ($k_{death}$ and $k_{stasis}$) by normalizing each metric measured for the heterogeneous population to that in a homogeneous population (i.e. $\omega = 0$ or $r = 1$) at different times of treatment. Smaller resistance enrichment ratios represent greater sensitivity to the presence of heterogeneous drug-tolerant cells.

We first compared the ability of each metric to capture the presence of small subpopulations of drug-tolerant cells by analyzing how resistance enrichment ratio varies with $\omega$ and time (Fig 2B and Supporting Information S1A and S1B Fig). Simulation results show that $f_a$ metrics, defined based on either normalized cell viability or growth rate inhibition (GR and DIP), are significantly less sensitive than $k_{death}$ and $k_{stasis}$ to the presence of drug-tolerant cells. Furthermore, for any given initial fraction of drug-tolerant cells ($\omega$), phenotype rate constants captured the emergence of drug resistance at earlier timepoints. Using similar simulations, we also tested how the relative level of drug resistance ($r$) in a fixed initial fraction of drug-tolerant cells would influence each of the drug response metrics. Simulation results show that for a

given ω, phenotype rate constants detect subpopulations with weaker levels of resistance (i.e. smaller $r$ values) and at earlier timepoints (Fig 2C and Supporting Information S1C and S1D Fig).

Most conventional drug screening assays are performed following exposure of cells to drug for 3 to 5 days. While variations in drug potency are distinguishable in such assays, it is often suggested that longer periods of treatment are essential to detect the phenotypic consequences of drug-tolerant persisters that diminish the efficacy. However, our results show that phenotype rate constants can capture heterogeneities that would otherwise require significantly longer experiments when using population-level $f_a$ metrics that mask such heterogeneities. The benefit of using phenotype rate constants would be especially significant in the case of potent drugs that induce substantial cell death, while sparing a small fraction of drug-tolerant cells (Supporting Information S2A and S2B Fig). In particular, as the efficiency of drug-induced cell killing increases, the sensitivity of $f_a$ metrics to detect drug tolerance in the surviving fraction of cells decreases (Supporting Information S2C Fig).

### Estimating probabilistic rate constants using time-lapse live cell microscopy

To experimentally capture stochastic processes of induction of cell death and inhibition of division in drug-treated tumor cell populations, we used time-lapse live cell microscopy and cells engineered to express two fluorescent reporters. The reporters included: (i) an H2B-Venus reporter which labels chromatin, allowing identification of nuclei and scoring cell death based on changes in nucleus morphology, and (ii) an mCherry-Geminin reporter for cell cycle progression [39] which allows tracking of cell division events. Using a high-throughput, automated image analysis workflow (see Materials and methods), the occurrence of individual phenotypic events (death and division) in single cells was tracked in time across a variety of drug treatment conditions (Supporting Information S3 Fig). To estimate time-dependent changes in probabilistic phenotype rate constants, the number of cell death and division events ($N_{event}$) were quantified over a series of uniform time intervals of length $\Delta t$. Phenotype rate constants were then estimated via normalizing $N_{event}$ in each time interval to the length ($\Delta t$) and the average number of live cells over that time interval $[N_{live}(t \rightarrow t + \Delta t)]_{avg}$ as detailed below:

$$k_{death}(t) = \frac{N_{death}(t \rightarrow t + \Delta t)}{[N_{live}(t \rightarrow t + \Delta t)]_{avg}\Delta t} \tag{5}$$

$$k_{division}(t) = \frac{N_{division}(t \rightarrow t + \Delta t)}{[N_{live}(t \rightarrow t + \Delta t)]_{avg}\Delta t} \tag{6}$$

$$k_{stasis}(t) = k_{division(no\ drug)} - k_{division(with\ drug)}(t) \tag{7}$$

As a proof of concept, we monitored responses of two BRAF-mutated melanoma cell lines (COLO858 and MMACSF) following exposure to a BRAF inhibitor Vemurafenib at 6 doses for a period of ~120 h. Heterogeneity in drug response was then visualized through the estimation and analysis of phenotype rate constants, $k_{death}$ and $k_{stasis}$, as a function of drug dose and time in each cell line (Fig 3A and 3B). In COLO858 cells, which have been shown to be initially sensitive but rapidly develop adaptive resistance to Vemurafenib [21,22], increasing drug concentration enhanced both the amplitude and the rate of increase in $k_{death}$ and $k_{stasis}$ within the first 36 h. After that, these responses were attenuated concurrent with the activation of drug-induced adaptive responses (Fig 3A). Responses of MMACSF cells involved a relatively

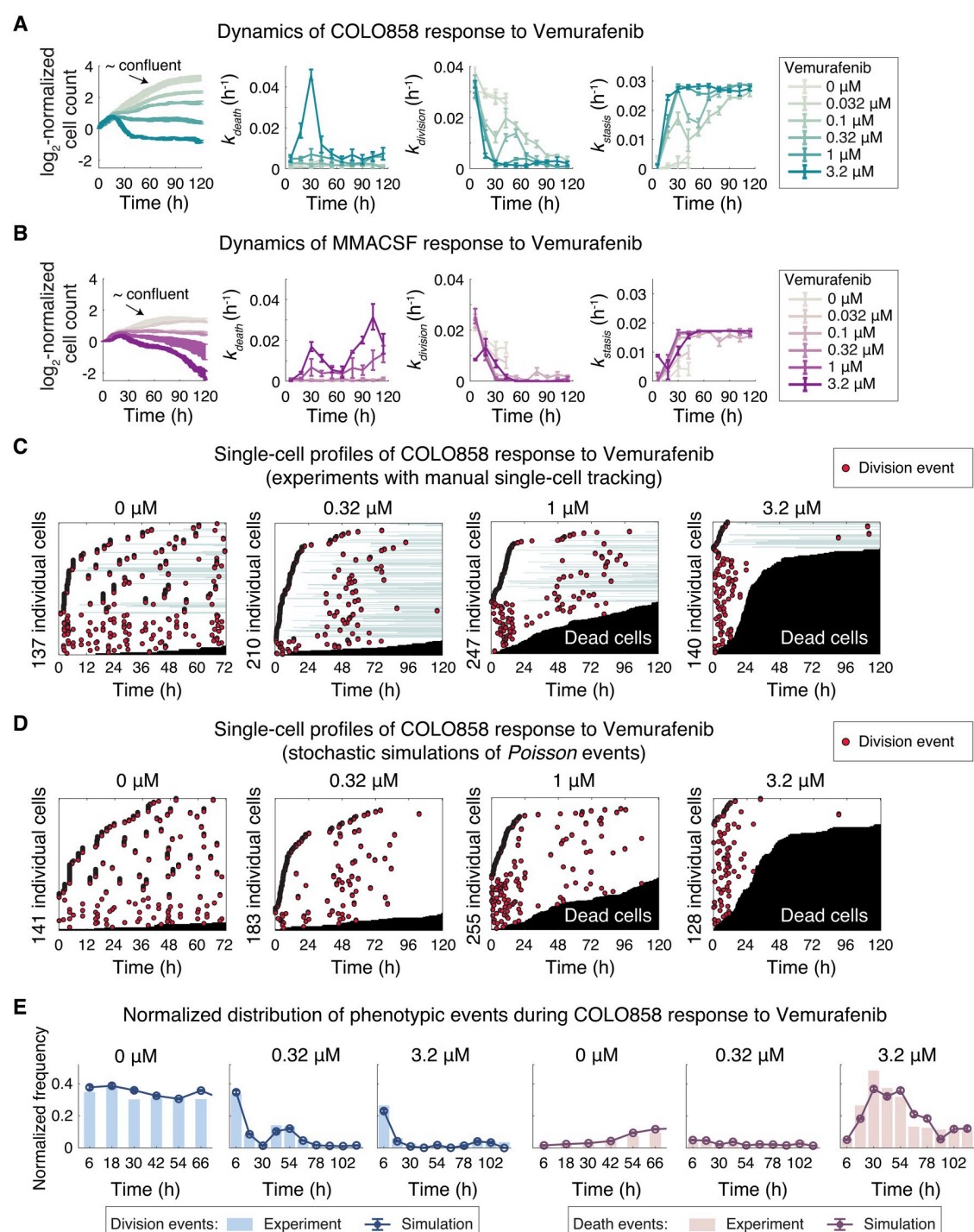

**Fig 3. Dynamic analysis of heterogeneous drug response using estimates of probabilistic phenotype rate constants from time-lapse live cell microscopy. (A-B)** Dynamics of (A) COLO858 and (B) MMACSF cell responses to BRAF inhibitor Vemurafenib across 6 doses (0–3.2 μM). Time- and dose-dependent changes in live cell count and estimates of $k_{death}$, $k_{division}$ and $k_{stasis}$ for time intervals of $\Delta t$ = 12 h are shown. Experimental data for Vemurafenib concentrations of 0 and 0.032 μM are shown until 48 h, that is when cells reached confluency under these conditions. Data are shown as mean ± SEM across four replicates. **(C)** Single-cell profiles of COLO858 response to Vemurafenib depicted based on manual tracking of individual cells exposed to different concentrations of Vemurafenib as described in (A). Each cell track is presented horizontally along time axis. Division events are marked as red circles. Transition from white to black represents a cell death event. Times at which cells spend out of field of view are shown in light green. **(D)** Single-cell profiles of COLO858 response to Vemurafenib simulated based on *Poisson* processes using rate parameters estimated

from COLO858 experimental data along 12 h time intervals. **(E)** Comparison of normalized distributions of division and death events along 12 h time intervals between experiments performed in COLO858 cells and the simulated responses for the same conditions. Experimental data-points represent pooled data from all four replicates. Simulated data-points represent mean ± SEM across 30 simulations.

monotonic and dose-dependent decrease in the number of live cells. At the highest drug concentration (3.2 μM), however, we observed two peaks of apoptotic response, one similar to COLO858 cells at $t \approx 36$ h and a higher peak later at $t \approx 108$ h (Fig 3B). These data are consistent with previous data reporting high sensitivity of MMACSF cells to 5 days of exposure to Vemurafenib [21,22], but also highlight the impact of cell-to-cell heterogeneity and the presence of subpopulations of cells with different levels of drug tolerance.

In addition to interrogating dynamic aspects of heterogeneous drug response, we also tested the performance of our automated image analysis workflow by comparing the estimated phenotype rate constants with those measured from data generated by manual single-cell tracking using a MATLAB-based software [40]. The software allowed accurate tracking and cell fate annotation of individual cells across time-lapse images taken over a period of multiple days. Single-cell profiles from manual tracking confirmed heterogeneity in the number and timing of death and division events in cells exposed to drug. In COLO858, for example, cell-to-cell variability ranged from cells that died rapidly (as early as ~24 h) in response to high concentrations of Vemurafenib, to cells that survived but did not divide, to cells that slowly divided following a temporary delay in their cell cycle, the proportion and dynamics of which changed with drug dose (Fig 3C). By comparing rate constants between two image analysis methods across a variety of conditions in two cell lines, we identified quantitatively similar patterns (Supporting Information S4 Fig). This consistency confirms that the automated workflow would be suitable for high-throughput analysis of drug response.

We also used single-cell phenotype data to empirically evaluate the assumption of non-stationary *Poisson* process to model drug-induced death and division events. We compared the distribution of phenotypic events measured from time-lapse microscopy experiments with those simulated based on *Poisson* processes using estimates of phenotype rate constants. We observed similarity across patterns of response at the single-cell level and between distributions of events at the population level (Fig 3C–3E), suggesting that a simplified model of non-stationary *Poisson* process for drug-induced death and division events is a reasonable one.

Taken together, high-throughput estimation and analysis of phenotype rate constants and their changes with time and dose provide an efficient tool to capture critical dynamic aspects of probabilistic and heterogeneous drug response that would be overlooked in bulk population assays.

## Evaluating statistical independence of drug combination efficacies using probabilistic phenotype metrics

Among the most widely used reference models in evaluation of synergistic efficacy for cancer drug combinations is Bliss independence [16]. The Bliss model assumes that drug effects are consequences of probabilistic processes, and that two drugs act independently if their combined effect confers probabilistic or statistical independence:

$$P^I_{A+B} = P_A + P_B - P_A P_B \qquad (8)$$

where $P^I_{A+B}$ describes the expected probability of the combinatorial effect of drugs A and B when they act independently. $0 \le P_A \le 1$ and $0 \le P_B \le 1$ represent probabilities of effect mediated by drugs A and B when tested individually. The Bliss combination index (*CI*) for drugs A

and B is defined as:

$$CI^{Bliss}_{A+B} = \frac{P^I_{A+B}}{P_{A+B}} \tag{9}$$

where $P_{A+B}$ describes the actual probability of effect induced by drugs A and B when used in combination. Synergistic combination efficacy is concluded if $CI < 1$, i.e. when the observed combinatorial effect exceeds the expected effect from the Bliss independence model. Despite its probabilistic definition, however, the Bliss model is broadly applied to a variety of $f_a$ metrics (such as normalized viability or net growth rate inhibition), thereby leading to unreliable conclusions which are largely due to the following limitations. First, although $f_a$ measurements satisfy the mathematical requirement of $0 \leq f_a \leq 1$, they do not have a probabilistic nature and thus do not necessarily follow the rules of probability theory. Second, $f_a$ quantities are the result of two distinct probabilistic processes, induction of cell death and inhibition of cell division. These processes, even when induced by drugs with the same probabilities, do not necessarily have the same impact on $f_a$. Third, for drugs A and B with $f_a < 1$, the Bliss model (when applied to $f_a$) is unable to account for the difference between being affected by drug A, drug B, or both. For example, consider the combined effect of two truly independent and purely cytostatic drugs A and B, whose phenotypic effects (individually) are described by $P_{stasis} = 1$ (and $P_{death} = 0$). By Bliss independence when applied to $f_a$ metrics such as viability or normalized growth rate inhibition, the combination of drugs A and B is expected to exhibit substantial cytotoxic effect and thus their combination would be scored incorrectly as antagonistic ($CI > 1$) (Fig 4A and 4B). To overcome such limitations and to avoid erroneous conclusions about drug combination efficacies, we propose to use probabilities of phenotypic events or their associated rate constants in evaluation of Bliss independence according to its probabilistic definition.

When applied to probabilistic events of drug-induced cell death, Bliss independence for the combined cytotoxic effect of drugs A and B is described as follows:

$$P^I_{death(A+B)} = k^I_{death(A+B)}dt = P_{death(A)} + P_{death(B)} - P_{death(A)}P_{death(B)} \approx (k_{death(A)} + k_{death(B)})dt \tag{10}$$

where $P_{death\ (A)}$ and $P_{death\ (B)}$ represent the probabilities with which drugs A and B induce cell death within a short time interval of $dt$, respectively. $P^I_{death\ (A+B)}$ represents the probability of death induced by the combination of drugs A and B when they act independently. $k_{death\ (A)}$, $k_{death\ (B)}$ and $k^I_{death\ (A+B)}$ represent rate constants associated with these probabilistic events, respectively. When applied to the conditional event of inhibition of cell division given that cells divide at a rate of $k_{division\ (no\ drug)}$ in the absence of drug, Bliss independence is described as follows:

$$P^I_{stasis(A+B)} = \frac{k^I_{stasis(A+B)}}{k_{division(no\ drug)}} = P_{stasis(A)} + P_{stasis(B)} - P_{stasis(A)}P_{stasis(B)} \tag{11}$$

where $P_{stasis\ (A)}$ and $P_{stasis\ (B)}$ represent the probabilities with which drugs A and B inhibit cell division given that cells would divide with a probability of $P_{division\ (no\ drug)} = k_{division\ (no\ drug)}dt$ within a short time interval of $dt$. $P^I_{stasis\ (A+B)}$ represents the cytostatic effect for the combination of drugs A and B when they act independently. $k_{stasis\ (A)}$, $k_{stasis\ (B)}$ and $k^I_{stasis\ (A+B)}$ represent rate constants associated with these probabilistic events. The Bliss combination index (for each

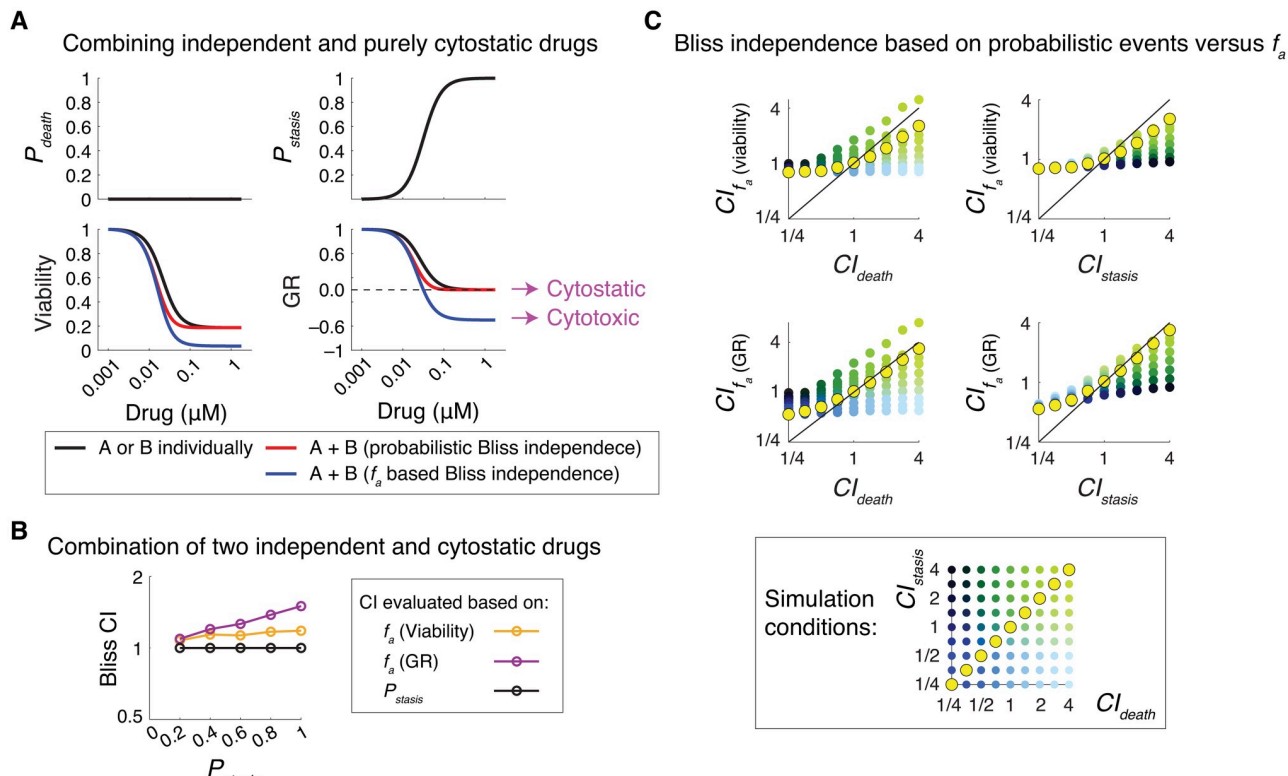

**Fig 4. Probabilistic phenotype metrics, but not $f_a$ based metrics, reveal statistical independence in drug combination efficacies. (A)** Simulation results shown for the effect of two independent and purely cytostatic drugs, A and B, with identical dose-effect profiles used individually and in combination. Dose-effect profiles of drugs A and B are shown as $P_{death}$ and $P_{stasis}$, quantified per unit of time (h) and per devision event, respectively. Normalized changes in relative viability and growth rate (GR) inhibition are reported for each condition at 48 h and the predicted combination effects are shown for scenarios where either the probabilistic metric $P_{death}$ and $P_{stasis}$ or $f_a$ quantities (based on viability and GR) where used in the evaluation of Bliss independence. **(B)** Bliss combination index values calculated (at t = 48 h) using different drug response metrics, $f_a$ (viability), $f_a$ (GR) and $P_{stasis}$, in simulations of combined effects of two independent and identical drugs with variable cytostatic effects represented by variations in $P_{satsis}$. The rate of cell division in the absence of drug was simulated as $k_{division\ (no\ drug)}$ = 0.035 h$^{-1}$. **(C)** Simulation results quantifying Bliss combination index values (at t = 48 h) calculated using $f_a$ response metrics, $f_a$ (viability) and $f_a$ (GR), in comparison with probabilistic combination index values ($CI_{death}$ and $CI_{stasis}$). Each data-point represents the mean of 10 simulations for a drug combination condition with a given set of probabilistic drug interaction condition quantified as $CI_{death}$ and $CI_{stasis}$. Conditions where $CI_{death} = CI_{stasis}$ are highlighted in yellow. Representative simulations were performed using an initial live cell number of $N_{live}$ (t = 0) = 2000, $k_{division\ (no\ drug)}$ = 0.035 h$^{-1}$, $k_{death\ (drug\ A)} = k_{death\ (drug\ B)}$ = 0.01 h$^{-1}$, $P_{stasis\ (drug\ A)} = P_{stasis\ (drug\ B)}$ = 0.2.

of the drug-induced phenotypic effects) is thus defined as follows:

$$CI_{death(A+B)}^{Bliss} = \frac{P_{death(A+B)}^{I}}{P_{death(A+B)}} = \frac{k_{death(A+B)}^{I}}{k_{death(A+B)}} \qquad (12)$$

$$CI_{stasis(A+B)}^{Bliss} = \frac{P_{stasis(A+B)}^{I}}{P_{stasis(A+B)}} = \frac{k_{stasis(A+B)}^{I}}{k_{stasis(A+B)}} \qquad (13)$$

Systematic simulation results show that evaluating probabilistic independence based on drug-induced phenotypic events can distinguish a variety of possible drug interactions that would be otherwise overlooked when assessed on the basis of $f_a$ quantities (Fig 4C). The discrepancy is particularly substantial under conditions where drug combinations have uneven cytotoxic and cytostatic interactions, e.g. when two compounds act synergistically with respect to inhibition of division but act independently or antagonistically with respect to induction of cell death, and vice versa.

## Probabilistic phenotype metrics uncover target-specific differences in drug combination efficacies

We applied the probabilistic definition of Bliss independence to evaluate time-dependent changes in the efficacies of a group of 12 compounds in sequential combination with BRAF kinase inhibitor, Vemurafenib, plus MEK kinase inhibitor, Trametinib, in two BRAF-mutated melanoma cell lines COLO858 and MMACSF over the course of five days (see Methods for details). Single-cell drug responses were monitored using time-lapse fluorescence microscopy, and changes in probabilistic rate constants $k_{stasis}$ and $k_{death}$ were tracked for the entire period of experiment for each drug condition individually or in combinations (Supporting Information S5 and S6 Figs). The list of compounds based on their nominal targets included two HDAC inhibitors (Fimepinostat and Givinostat), two BET bromodomain inhibitors (Birabresib and I-BET762), two KDM1A inhibitors (SP2509 and ORY-1001), a pan Jmj-KDM inhibitor (JIB-04), a KDM5 inhibitor (CPI-455), two Tankyrase inhibitors (AZ6102 and NVP-TNKS656), and two CDK4/6 inhibitors (Palbociclib and Abemaciclib). These compounds were selected from two broad classes of anti-cancer drugs, referred to as epigenetic-modifying compounds and cell cycle inhibitors, which have been proposed to be used in combination with standard of care BRAF and MEK inhibitors to overcome drug-adapted subpopulations of cells in BRAF-mutant melanomas [10,21,41–49].

The analysis of variations in Bliss combination index, defined based on probabilistic cytotoxic and cytostatic actions, with drug and time (following unsupervised clustering) led to two major conclusions (Fig 5A). First, effective drugs with comparable mechanisms of action (e.g. BET inhibitors, HDAC inhibitors or CDK4/6 inhibitors) exhibited similar dynamic patterns of interaction with BRAF and MEK kinase inhibitors, suggesting that differences in probabilistic drug action and interactions are target-specific. Second, cytostatic and cytotoxic drug interactions among efficacious drug combinations often varied in time and did not necessarily correlate with one another. BET inhibitors, for example, exhibited a strong synergistic cytotoxic interaction ($CI_{death} < 1$) with the combination of BRAF and MEK inhibitors within 48–72 h of treatment in both COLO858 and MMACSF cell lines, whereas their interaction was scored as independent ($CI_{stasis} \approx 1$) with respect to inhibition of cell division (Fig 5A and 5B). Furthermore, the benefit of BET inhibitors combined with Vemurafenib and Trametinib diminished following 96 h, concomitant with the emergence of a small proliferating subpopulation ($k_{division} > 0$) (Fig 5B). CDK4/6 inhibitors acted independently with BRAF and MEK inhibitor combination to inhibit cell division within 72–96 h in both cell lines (Fig 5A, Supporting Information S5 and S6 Figs). Surprisingly, however, their effects on cell death appeared to be antagonistic especially in MMACSF cells (Fig 5C). This might be due the possibility that upon G0/G1 arrest, BRAF-mutant cells become less responsive to the effect of BRAF and MEK inhibitors, an interesting observation which requires further investigation across more cell lines.

Altogether, experimental results and simulation outcomes suggest that dynamic measurements of the phenotype metrics and probabilistic evaluation of combination index reveal distinctive responses of cells to drug combinations that might be indistinguishable when assessed based on conventional assays. Phenotype metrics deconvolve differential (and sometimes opposing) degrees of drug effect on tumor cell killing versus inhibition of cell division, a potentially important consideration in choosing appropriate drug combinations. Furthermore, the probabilistic nature of these metrics makes them sensitive to cell-to-cell heterogeneities which are typically overlooked in conventional bulk drug response assays. They are therefore appropriate choices to assess synergistic efficacy in drug combinations aimed at blocking heterogeneous subpopulations of drug-tolerant cells.

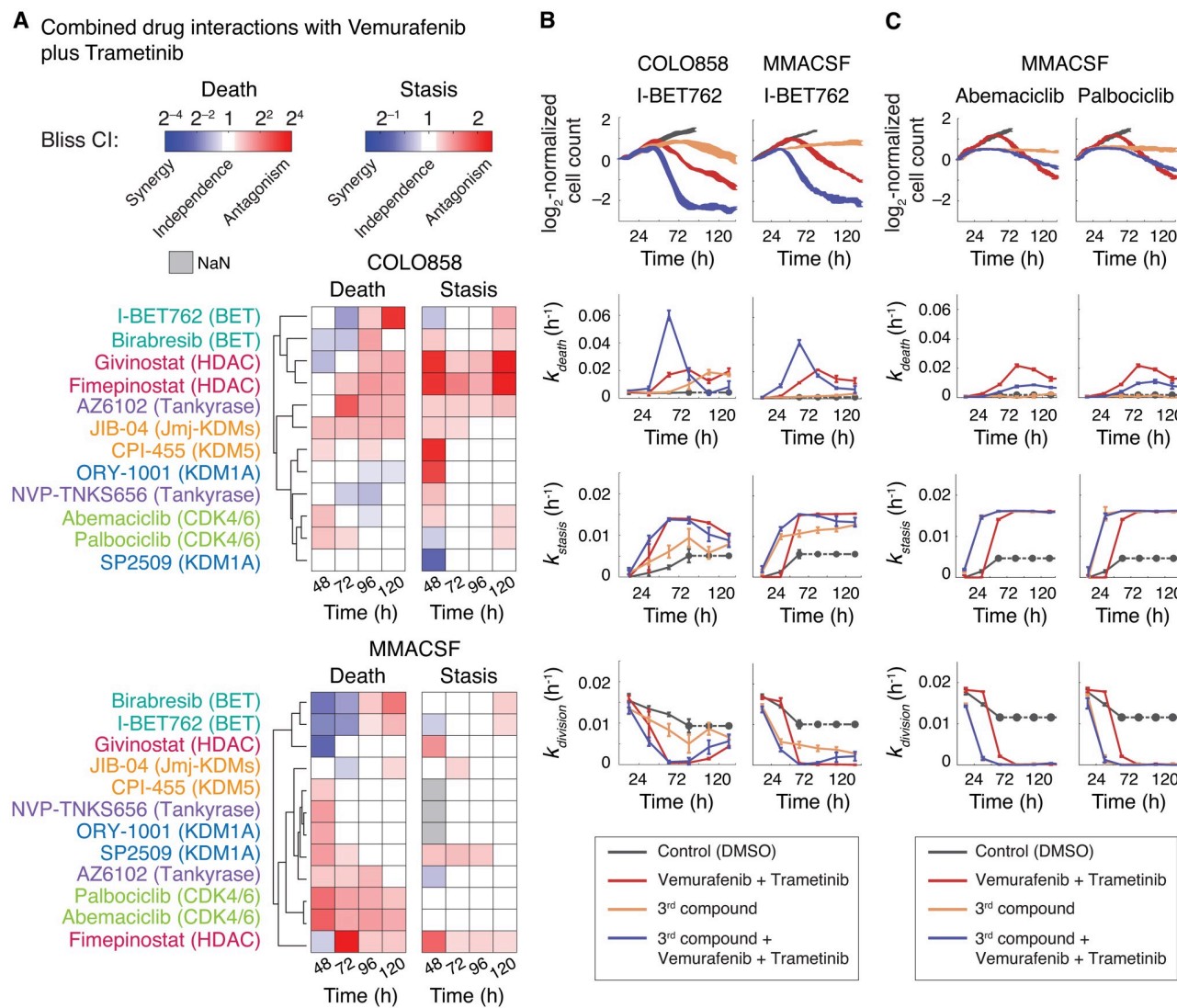

**Fig 5. Probabilistic phenotype metrics uncover target-specific differences in drug combination efficacies and their interactions. (A)** Unsupervised clustering of Bliss combination index values ($CI_{death}$ and $CI_{stasis}$) calculated using probabilistic metrics $k_{death}$ and $k_{stasis}$ in COLO858 and MMACSF cells between 48–120 h of exposure to various drugs in sequential combination with Vemurafenib and Trametinib. Cells were treated initially for 24 h with DMSO control or one of the epigenetic-modifying compounds or cell cycle inhibitors (3$^{rd}$ compound) at the following concentrations: Fimepinostat (0.02 μM), Givinostat (0.2 μM), Birabresib (0.5 μM), I-BET762 (1 μM), SP2509 (1 μM), ORY-1001 (1 μM), JIB-04 (0.2 μM), CPI-455 (5 μM), AZ6102 (1 μM), NVP-TNKS656 (1 μM), Palbociclib (1 μM), and Abemaciclib (1 μM). Nominal targets of compounds are highlighted. After 24 h, Vemurafenib at 0.3 μM plus Trametinib at 0.03 μM, or DMSO control were added to each treatment condition. Combination index data-points represent mean values across 2–3 replicates. NaN data-points represent conditions where the effect of drug combination or that of the independence model are within measurement error, making the ratio (combination index) unreliable. **(B)** Estimated dynamics of live cell count, $k_{death}$, $k_{stasis}$ and $k_{division}$ measured from time-lapse live cell microscopy data for COLO858 and MMACSF cell responses to the combination of Vemurafenib (0.32 μM) and Trametinib (0.032 μM), BET bromodomain inhibitor I-BET762 (1 μM), their triple combination, or vehicle (DMSO) control. **(C)** Estimated dynamics of live cell count, $k_{death}$, $k_{stasis}$ and $k_{division}$ measured from time-lapse live cell microscopy data for MMACSF cell responses to the combination of Vemurafenib (0.32 μM) and Trametinib (0.032 μM), CDK4/6 inhibitors Palbociclib (1 μM) and Abemaciclib (1 μM), their triple combination, or vehicle (DMSO) control. $k_{division\ (no\ drug)}$ used for the estimation of $k_{stasis}$ for each cell line was estimated using cell division data averaged for the first 24 h in cells treated with DMSO only. In conditions where confluency was achieved (e.g. DMSO-treated cells after 60 h), data-points were replaced with the last available data-point (dotted line). Data-points represent mean ± SEM across 2 or 3 replicates.

## Discussion

Synergistic interactions in cancer drug efficacy are typically assessed using Bliss independence or other models (e.g. Highest Single Agent approach [2]). These models are commonly applied

to drug response measurements, whose outcomes are normalized to those measured in untreated controls to identify the fraction of cells affected ($f_a$) by drugs. Examples of these metrics include drug-induced changes in viability (normalized live cell count) or net growth rate inhibition, which are analyzed across drug doses and combinations. Synergistic efficacy is then concluded when the observed combinatorial effect on $f_a$ metrics exceeds the expected effect from the null model. In this paper, we use basic probability theory and computer simulations to demonstrate that using $f_a$ metrics may bias our estimation of drug combination effectiveness and synergistic efficacy, especially when the ultimate goal is to block heterogeneous drug-tolerant subpopulations. Instead, we propose to use direct measures of time-dependent probabilities of key drug-induced phenotypic events, i.e. induction of cell death and inhibition of cell division, and their associated rate constants ($k_{death}$ and $k_{stasis}$) to evaluate synergistic efficacy using probabilistic models such as Bliss.

Probabilistic phenotype metrics improve our ability to quantify drug efficacy and characterize drug combination interactions in the following three ways. First, in contrast to the commonly used $f_a$ metrics, phenotype metrics are directly related to the probabilities of drug action in a cell population within any given time interval following drug exposure. Furthermore, they deconvolve differential degrees of drug effect on tumor cell killing versus inhibition of cell division, which may not be correlated in many cases. Second, $k_{death}$ and $k_{stasis}$ dramatically increase the sensitivity of short-term drug response assays to dynamic cell-to-cell heterogeneities and the presence (or emergence) of drug-resistant sub-clones, which are typically overlooked in conventional $f_a$ based drug response analyses. This is a critical issue especially when heterogeneous tumor cell populations consist of cells that are differentially sensitive to drugs and that their sensitivity changes with time. Third, the probabilistic nature of phenotype metrics allows us to use them directly in unbiased evaluation of independence, synergistic or antagonistic efficacy in drug combinations using probabilistic models such as Bliss independence.

While we focused on Bliss independence as a probabilistic framework to study synergistic efficacy, phenotype metrics and their dose- and time-dependent variations could be used in other platforms for broad evaluation of synergy. A recently developed multi-dimensional framework (MuSyC) uses a two-dimensional extension of Hill equation to distinguish synergistic efficacy versus synergistic potency, thereby allowing for a comprehensive understanding of drug interactions. Such understanding not only helps with improving therapeutic efficacy via enhancing effect, but also reducing off-target toxicities via dose reduction [10]. Probabilistic phenotype rate constants follow dose-response patterns suitable to be fit by Hill equation and therefore can be used as input to platforms such as MuSyC.

Estimating probabilistic phenotype metrics requires continuous time-lapse experiments along periods of multiple days, followed by computational single-cell analysis. In this study, we used cell lines engineered to express fluorescent reporters to capture drug-induced changes in cellular death and division events. Such integrative methods may not be necessary for large-scale drug screening projects, in which many drugs are filtered out because of lack of potency. The benefit of these methods is significant, however, when there is a need for identifying more efficacious drugs or drug combinations among a selection of reasonably potent candidates. It has become increasingly evident that cell-to-cell variability is the cause of partial efficacy and incomplete responsiveness of tumor cell populations to a variety of highly potent cytotoxic and targeted therapies. Such heterogeneities are not captured using conventional population-based assays, but may originate residual cells from which drug-resistant clones can arise. Therefore, the probabilistic analysis of single-cell phenotypes has a great potential to improve our understanding of heterogeneity in drug response and facilitate the discovery of more efficacious combination therapies. We envision that the ongoing experimental and computational

advances in single-cell tracking (including dye-based or label-free cell fate tracking and lineage construction by using machine learning and deep learning algorithms) will rapidly improve the efficiency, accuracy and applicability of single-cell approaches to the analysis of cancer drug response.

## Materials and methods

### Cell culture

BRAF-mutated melanoma cell lines used in this study were obtained from the Massachusetts General Hospital Cancer Center with the following primary sources: COLO858 (ECACC) and MMACSF (Riken Bioresource Center). Each cell line was independently authenticated by Short Tandem Repeats (STR) profiling by ATCC. COLO858 cells were grown in RMPI 1640 (Corning cellgro, Cat. 10–040 CV), and MMACSF cells were grown in DMEM/F-12 (Thermo Fisher Scientific, Cat. 11330–032). For both cell lines, growth media were supplemented with 5% fetal bovine serum (Thermo Fisher Scientific, Cat. 26140–079) and 1% sodium pyruvate (Thermo Fisher Scientific, Cat. 11360–070). We added penicillin and streptomycin at 100 U/ml (Thermo Fisher Scientific, Cat. 15140–122) and plasmocin at 0.5 μg/ml (InvivoGen, Cat. ant-mpp) to all growth media. Cells were engineered to stably express H2B-Venus and mCherry-Geminin fluorescent reporters as described previously [21]. Engineered and parental cell lines were confirmed to grow at comparable rates in the absence of any treatment or in the presence of different concentrations of BRAF inhibitor Vemurafenib over 72 hours of treatment.

### Reagents

Chemical inhibitors used in this study were obtained from Selleck Chemicals with the following catalog numbers: Vemurafenib (Cat. S1267), Trametinib (Cat. S2673), SP2509 (Cat. S7680), ORY-1001 (Cat. S7795), Palbociclib (Cat. S1116), Abemaciclib (Cat. S7158), AZ6102 (Cat. S7767), NVP-TNKS656 (Cat. S7238), Givinostat (Cat. S2170), Fimepinostat (CUDC-907; Cat. S2759), JIB-04 (Cat. S7281), CPI-455 (Cat. S8287), I-BET762 (Cat. S7189) and Birabresib (OTX-015; Cat. S7360). All chemical inhibitors were dissolved in dimethyl sulfoxide (DMSO) as 10 mM stock solution, except Palbociclib of which the stock concentration was 5 mM.

### Cell seeding and drug treatment

COLO858 and MMACSF cells expressing fluorescent reporters were seeded into Costar 96-well black clear-bottom tissue culture plates (Corning 2603) in 220 μL full growth medium without phenol red at a density of 2000 and 3000 cells/well, respectively. Cells were counted using a TC20 Automated Cell Counter (Bio Rad). In the case of Vemurafenib dose-response experiments, cells were treated ~24 h after seeding with either DMSO or five different concentrations of Vemurafenib (0.0316, 0.1, 0.316, 1 and 3.16 μM) for a period of ~5 days. In the case of drug combination experiments, cells were treated (also 24 h after seeding) with DMSO control or one of the epigenetic-modifying compounds or cell cycle inhibitors at the following concentrations: Fimepinostat (0.02 μM), Givinostat (0.2 μM), Birabresib (0.5 μM), I-BET762 (1 μM), SP2509 (1 μM), ORY-1001 (1 μM), JIB-04 (0.2 μM), CPI-455 (5 μM), AZ6102 (1 μM), NVP-TNKS656 (1 μM), Palbociclib (1 μM), and Abemaciclib (1 μM); drug concentrations were chosen based on previous reports exhibiting maximal target inhibition in cells. After 24 h, Vemurafenib at 0.3 μM plus Trametinib at 0.03 μM, or DMSO control were added to each treatment condition. All drug treatments were performed in at least 4 replicates using a Hewlett-Packard (HP) D300 Digital Dispenser.

## High-throughput time-lapse live cell microscopy

Within 50–60 min after each treatment, cells were imaged every 10 min (for COLO858) and every 15 min (for MMACSF) using a Nikon Ti2-E inverted microscope with motorized stage, Perfect Focus System, 20× objective, and a Photometrics Prime 95B camera followed by 2×2 binning. The process of image acquisition was controlled using NIS element software. Illumination was powered by the Lumencor Spectra X light engine. H2B-Venus fluorescence was captured using 510 nm excitation and 535 nm emission at 25 ms exposure for MMACSF cells and 20 ms exposure for COLO858 cells. mCherry-Geminin fluorescence was captured using 575 nm excitation and a 629.5 nm emission at 80 ms exposure for MMACSF cells and 100 ms exposure for COLO858 cells. Throughout the entire period of image acquisition, environmental conditions were maintained at 37˚C, 5% $CO_2$, and 93% humidity using an OkoLab Enclosure.

## Image analysis and automated cell tracking workflow

Images were first processed using Fiji [50] for rolling ball background subtraction with radius of 20 pixels. Background-subtracted images were then analyzed using CellProfiler (3.1.8) for segmentation and classification of cellular phenotypic states, including cells that express high and low levels of Geminin (referred to as Geminin$^{high}$ and Geminin$^{low}$ cells, respectively), or live versus dead cells. Briefly, CellProfiler analysis (Supporting Information S3 Fig) involved: (1) edge enhancement and dark hole feature enhancement of the background-subtracted H2B images to facilitate segmentation; (2) segmenting individual cell nuclei using the Otsu thresholding method; (3) using nuclei segmentations as masks to measure object intensities for all channels as well as object sizes and shapes; (4) classification of phenotypic states of each cell object using the classification model output from CellProfiler Analyst (2.2.1) [51] based on features measured from the previous step. Geminin$^{high}$ versus Geminin$^{low}$ cell classifiers and live versus dead cell classifiers were trained separately using fast gentle boosting algorithm in CellProfiler Analyst with eight and fifteen maximum rules, respectively. The training set used to develop each phenotype classifier was an annotated set, generated via manually sorting the cell object tiles into their corresponding phenotype classes in CellProfiler Analyst. The process of manual sorting followed by model training was iterated until approximately 80% of true positive accuracy was achieved.

Based on phenotype classifications of individual cells for each image output from CellProfiler, corresponding synthetic images were generated in MATLAB (2018b) for each phenotype of interest. Synthetic images contained synthetic pixels at locations of cells. They were used to mark each of the phenotypes of interest (i.e. Geminin$^{high}$ or dead cells) in each image, which were then tracked across series of timepoints using the Fiji plugin TrackMate (3.8.0), without additional need to perform image segmentation. In other words, we combined the power of CellProfiler analysis for accurate and high-throughput image segmentation with the ability of TrackMate to track individual cell phenotypes in a multi-day time-lapse experiment. To achieve this goal, synthetic images for Geminin$^{high}$ cells, dead cells, background-subtracted H2B and Geminin images acquired from the same site were merged into a single multi-channel image composite using Fiji. Image composites were then analyzed using TrackMate with TrackMate extras and Track Analysis extensions [52] for automated tracking. Synthetic pixels of a selected channel were detected by the Laplacian of Gaussian detector and spots were linked with Linear Assignment Problem (LAP) Tracker. Additional spots filtering (based on intensities from multiple channels) and track filtering (based on track duration, track median velocity, and velocity standard deviation) were implemented to optimize tracking results.

### Estimating probabilistic phenotype rate constants from individual cell tracking data

Single-cell tracking data generated using TrackMate was analyzed using MATLAB. Transition of a live cell from Geminin$^{high}$ to Geminin$^{low}$ was recorded as a division event, whereas the beginning of a dead cell track was recorded as a death event (Supporting Information S3 Fig). To estimate time-dependent changes in probabilistic phenotype rate constants, $k_{death}$ and $k_{division}$, the number of recorded cell death and division events ($N_{death}$ and $N_{division}$) were quantified over a series of uniformly distributed time intervals ($t \rightarrow t + \Delta t$), where $\Delta t = 12$ h or 24 h. Normalizing $N_{death}$ and $N_{division}$ to the length of each time interval ($\Delta t$) and the average number of live cells within the same interval $[N_{live}(t \rightarrow t + \Delta t)]_{avg}$, phenotype rate constant were estimated using Eqs 5–7. As expected, we observed that the magnitude of noise in single-cell tracking data and consequently the relative error in the estimation of $k_{division}$ and $k_{death}$, increased under conditions where Geminin$^{high}$ and dead cells were highly concentrated, respectively. To mitigate the effect of noise under such conditions, we imposed the following constraints in our estimation of $N_{division}$ (when $N_{death} < N_{division}$) and $N_{death}$ (when $N_{death} > N_{division}$) during each time interval, respectively:

$$N_{division}(t \rightarrow t + \Delta t) = (N_{live}(t + \Delta t) - N_{live}(t)) - N_{death}(t \rightarrow t + \Delta t) \tag{14}$$

$$N_{death}(t \rightarrow t + \Delta t) = N_{division}(t \rightarrow t + \Delta t) - (N_{live}(t + \Delta t) - N_{live}(t)) \tag{15}$$

These constraints are consistent with the assumption that the overall change in the number of live cells during each time interval ($\Delta t$) must be equal to the number of division events minus the number of death events during the same time interval.

### Verifying the accuracy of automated cell tracking workflow using manual single-cell tracking

To test the performance of our automated image analysis workflow, we compared the phenotype rate constants measured using data from the automated pipeline with those measured using data generated from manual single-cell tracking. This was accomplished using a MATLAB-based software, allowing accurate single-cell tracking and cell fate annotation of individual cells across time-lapse images taken over a period of multiple days [40]. Briefly, the manual tracking method relies on identification of individual cells using intensity and shape information of the nuclear marker (H2B-Venus), track propagation using nearest neighbor criteria, and real-time user correction of tracking, and annotation of cell death and division events based on H2B and Geminin signal intensities. For each condition, about 150–250 cells pooled from four replicates were manually tracked and cell death and division events were recorded. Phenotype rate constants were then calculated using Eqs 5–7.

### Estimating fraction of cells affected ($f_a$) by drug

Currently, evaluation of Bliss independence (and other drug interaction frameworks) is based on fraction of cells affected ($f_a$), a normalized parameter between zero and one, that represents the fractional effect of drugs individually or in combination [53]. Conventionally, relative viability (or cell count normalized to an untreated control) measured at a fixed time-point (typically 72 or 96 h) following drug treatment has been used to calculate $f_a$:

$$f_a(viability) = 1 - viability \tag{16}$$

Despite its wide-spread usage, however, the relative viability approach in assessing drug response suffers from a fundamental flaw, which is being confounded by variation in cell proliferation rates and assay duration. The reason is that cell count, which is used as a normalization factor in this approach, is non-linearly time-dependent. Therefore, new generation drug response metrics have recently been introduced to correct for this bias [5,6]. The nature of these metrics is based on modeling drug-induced changes in the net growth rate of the cancer cell population (instead of relative viability) as a function of drug dose. These metrics include drug-induced proliferation (DIP) rate [6] and growth rate (GR) inhibition [5], both of which consider and correct for the variability in growth rate that is irrelevant to drug treatment via normalizing the net growth rate of the drug-treated cell population to that of the untreated control. DIP rate and GR inhibition are defined as follows:

$$DIP = \frac{k_{net\ growth(with\ drug)}}{k_{net\ growth(no\ drug)}} \tag{17}$$

$$GR = 2^{\frac{k_{net\ growth(with\ drug)}}{k_{net\ growth(no\ drug)}}} - 1 \tag{18}$$

where $k_{net\ growth\ (with\ drug)}$ and $k_{net\ growth\ (no\ drug)}$ are the net population growth rates measured in the drug-treated cell population and the untreated control at a particular time-point, respectively.

Given the dynamic range of each metric, the definition of fraction of cells affected ($f_a$) for these new metrics, $f_a$(DIP) and $f_a$(GR), is modified as follows so that $0 \leq f_a \leq 1$:

$$f_a(DIP) = \frac{1 - DIP}{1 - \min(DIP)} \tag{19}$$

$$f_a(GR) = \frac{1 - GR}{2} \tag{20}$$

## Stochastic simulation of cytotoxic and cytostatic drug effects

We modeled phenotypic events in a heterogeneous tumor cell population as a series of independent stochastic reaction processes at a single-cell level. Drug-induced death events were described by the following reaction:

$$cell \xrightarrow{k_{death}} dead\ cell$$

where the rate constant of death $k_{death}$ is defined such that a given cell dies with a probability of $k_{death}dt$ within a reasonably short time interval ($dt$). Cell division in the absence of drug may be described by the following reaction:

$$cell \xrightarrow{k_{division(no\ drug)}} 2cells$$

where $k_{division\ (no\ drug)}$ is the inherent rate of division of a given cell. The cytostatic effect of a drug on a cell was described by a conditional probability ($P_{stasis} = P_{inhibition\ of\ division\ (with\ drug)\ |\ division\ (no\ drug)}$) with which it prevents a cell from dividing given that it would have divided in the absence of drug with a probability of $P_{division\ (no\ drug)}$. Drugs that do not inhibit cell division and those that accelerate cell division are both characterized by $P_{stasis} = 0$. To satisfy this assumption, we consider an upper-bound limit for the probability of cell division that is equal to $P_{division\ (no\ drug)}$.

For cancer drugs that have inhibitory effect on cell division, the relationship between $P_{stasis}$ and $P_{division\ (with\ drug)}$ may be derived as follows:

$$P_{division(no\ drug)} = P_{division(no\ drug)\cap division(with\ drug)} + P_{division(no\ drug)\cap inhibition\ of\ division(with\ drug)} \tag{21}$$

$$P_{division(no\ drug)} = P_{division(with\ drug)} + P_{inhibition\ of\ division(with\ drug)|division(no\ drug)} \cdot P_{division(no\ drug)} \tag{22}$$

$$P_{inhibition\ of\ division(with\ drug)|division(no\ drug)} = \frac{P_{division(no\ drug)} - P_{division(with\ drug)}}{P_{division(no\ drug)}} \tag{23}$$

$$P_{stasis} = 1 - \frac{P_{division(with\ drug)}}{P_{division(no\ drug)}} \tag{24}$$

$$P_{division(with\ drug)} = (1 - P_{stasis})P_{division(no\ drug)} \tag{25}$$

In the presence of drug, cell division and inhibition of cell division (stasis) may be described by the following reactions, respectively:

$$cell \xrightarrow{k_{division}} 2\ cell$$

$$cell \xrightarrow{k_{stasis}} cell$$

where the rate constants are as follow:

$$k_{stasis} = P_{stasis}k_{division(no\ drug)} \tag{26}$$

$$k_{division} = (1 - P_{stasis})k_{division(no\ drug)} \tag{27}$$

The model assumes that the processes of drug-induced cell death and inhibition of cell division are independent of each other.

At the population level, *Poisson* processes of drug-induced phenotypic events in a tumor cell population were simulated using the Gillespie algorithm. Briefly, the Gillespie algorithm determines the time to the next reaction event in the cell population based on an exponential distribution that statistically characterizes the *Poisson* processes. The algorithm then stochastically determines whether the event is death or division based on probabilities that are proportional to the rates of these two processes ($k_{death}$ and $k_{division}$). If the chosen event is division, then with probability $P_{stasis}$ that division event is rejected.

## Validation of non-stationary *Poisson* models for live cell microscopy data

The probabilistic modeling approach used in this study involves a commonly used formulation of stochastic chemical kinetics, describing the time evolution of a reacting system while taking into consideration the fact that individual reaction events (typically between individual molecules) are random point *Poisson* processes [54]. While using a similar formulation, we consider a single-cell event (death or division) as an individual event, rather than modeling any of the drug-induced molecular events that underlie such cellular processes. Previous work (based on live-cell measurement and modeling) has shown that apoptosis, for example, is controlled by a snap action switch at a single-cell level [55]. This switch, however, is associated with a delay that is variable from one cell to another within a population, preventing cells

from dying en masse following exposure to a death stimulus. Such variability makes it possible for an all-or-none response at a single-cell level to be graded at the population level such that the concentration of stimulus (or drug) controls the probability with which a cell responds during a specific period of time. This is in general agreement with the definition of *Poisson* process to model the population behavior, where the rate of a cellular event is directly linked to the *Poisson* rate parameter. Such parameter is expected to be controlled by the concentration of a key set of molecular regulators that are not explicitly modeled. However, changes in cellular state (due to drug adaptation, for example) may influence the concentration of these regulators, causing time-dependent changes in the *Poisson* rate constant, making non-stationary *Poisson* process a potentially suitable framework to model such dynamic, adaptive responses.

To experimentally test whether a simplified model of non-stationary *Poisson* process may explain the distribution of drug-induced death and division events in time-lapse microscopy data, we used maximum likelihood estimation to fit two non-stationary *Poisson* models, one to the single-cell death data and one to the single-cell division data. The rate function $k(t)$ of the non-stationary *Poisson* models used for data fitting was assumed to be a piece-wise function in time, where for each 12 h interval the rate was given by a single parameter. Hence, to capture 120 h of data, we set the rate function for each *Poisson* process with 10 parameters. The log-likelihood function for fitting a non-stationary *Poisson* model is given as follows:

$$l(\theta) = \sum_{j=1}^{n} \log\{k(t_j; \theta)\} - \int_0^T k(\tau; \theta)d\tau \qquad (28)$$

where $\theta$ is a vector of the 10 parameters to be estimated from the data, $n$ is the number of datapoints, $t_j$ is the time of the $j^{th}$ event and $T$ is the end time of the experiment. The log-likelihood function was then maximized using the constrained optimization function 'fmincon ()' in MATLAB. Using the fit parameters, we then simulated drug responses for 30 times and the normalized mean counts of phenotypic events were compared to that of the same data used for parameter estimation.

## Simulations of combined drug responses with variable modes of drug interaction

For combined drug response simulations, we modified the Gillespie algorithm as follows. After determining the time of the next event, the algorithm stochastically determines whether that event is a death event induced by drug A, a death event induced by drug B, or a division event based on probabilities proportional to their rates of occurrence. In cases where the two drugs confer statistically independent cell killing, the probabilities of the next event being drug A-induced death and drug B-induced death are respectively proportional to their single drug-induced rates of death, i.e. $k_{death(A)}$ and $k_{death(B)}$, whereas the probability of the next event being a division event is proportional to the inherent division rate of the cell, $k_{division\ (no\ drug)}$. If the next event is division, then with a probability of $P_{stasis(A+B)}$ that division event is rejected. For independent cytostatic interactions, $P_{stasis(A+B)}$ is set to $P^I_{stasis(A+B)}$ as defined in Eq 11. In cases where drug combinations are not independent, $P_{stasis(A+B)}$ and $P_{death(A+B)}$ will be calculated as $P^I_{stasis(A+B)}$ and $P^I_{death(A+B)}$ divided by $CI_{death}$ and $CI_{stasis}$ to simulate different modes of drug interaction, respectively. For the purpose of comparison, we also evaluated Bliss combination index while replacing probabilistic metrics with $f_a$ quantities measured for each drug condition individually and in combination.

## Simulations of heterogeneous drug response in the presence of drug-tolerant subpopulations

We simulated drug treatment scenarios where the initial cell population consisted of heterogeneous subpopulations, in which a small fraction of cells was substantially less sensitive to treatment relative to the majority of the cell population. Stochastic arrival of death and inhibition of division events were modeled using Gillespie algorithm as described above, while considering two subpopulations: a larger sensitive (S) subpopulation and a small drug-tolerant or resistant (R) subpopulation. We initialized simulations with 300 cells, a small fraction of which ($\omega$, varied from 0%-5%) had a more resistant phenotype, i.e. a lower death rate constant and a lower probability of stasis than that of the sensitive population, in the presence of drug. We modeled such resistant phenotype by defining the level of resistance ($r \geq 1$, varied from 1–16) as the fold-change in the rates of death and probability of stasis relative to the sensitive population. We used same fold-changes for death and cytostasis rates. We assumed a fixed inherent growth rate for the sensitive population $k^S_{division\ (no\ drug)} = 0.035\ \mathrm{h}^{-1}$, while considering three different possible inherent growth rates for the resistant population $k^R_{division\ (no\ drug)} = 0.035\ \mathrm{h}^{-1}$, $0.02\ \mathrm{h}^{-1}$, and $0.009\ \mathrm{h}^{-1}$. Drug response parameters for the drug-sensitive population include: $k^S_{death\ (drug)} = 0.03\ \mathrm{h}^{-1}$, $P^S_{stasis\ (drug)} = 0.8$. Drug response parameters for the drug-tolerant subpopulation are: $k^R_{death\ (drug)} = k^S_{death\ (drug)} / r$, $P^R_{stasis\ (drug)} = P^S_{stasis\ (drug)} / r$. We assumed that phenotypic responses of both subpopulations are independent of each other and that daughter cells within the same subpopulation inherit the exact same probabilities of phenotypic events as their mother cells. The responses (i.e. number of live cells, death and division events) of the two subpopulations were summed together to show the overall response of the entire cell population. To compare quantitatively the sensitivity of different metrics in capturing the differences in drug effect in the presence of phenotypic heterogeneity, we systematically varied the initial fraction of drug-tolerant subpopulation ($\omega$) and its level of resistance ($r$) as input parameters in simulations. For each simulation, overall drug effect using different metrics (fraction affected or phenotype rate constants) were calculated. To evaluate the sensitivity of each metric to the presence of drug-tolerant subpopulations, we defined and calculated "resistance enrichment ratio" as the ratio of these metrics between two treatment scenarios, one in the presence of a heterogeneous population (varying $\omega > 0$ and $r > 1$) and one in the absence of heterogeneity ($\omega = 0$ or $r = 1$). The smaller the resistance enrichment ratio becomes, the more significant decrease in drug effect is captured by a given metric in the presence of drug-tolerant cells.

## Hierarchical clustering

Unsupervised hierarchical clustering of combination index (*CI*) values estimated from the application of Bliss independence to probabilistic phenotype rate constants measured for 24 h time intervals of drug treatments was carried out using MATLAB 2018b with the Euclidean distance metric and the Complete (farthest distance) algorithm for computing the distance between clusters.

## Sensitivity analysis

To compare the sensitivity of different metrics (with different units), to the variation of parameter $P_{death}$, we defined fractional sensitivity ($S_f$) as follows:

$$S_f(f_a, P_{death}) = \frac{\partial f_a / f_a}{\partial P_{death} / P_{death}} \tag{29}$$

$$S_f(k_{death}, P_{death}) = \frac{\partial k_{death}/k_{death}}{\partial P_{death}/P_{death}} \tag{30}$$

$S_f$ values of less than 1 represent reduced sensitivity of the metric to changes in $P_{death}$.

## Statistical analysis

All data with error bars were presented as mean ± standard error of the mean (SEM) using indicated numbers of replicates.

## Code availability

Custom MATLAB scripts for probabilistic simulation of drug response in heterogeneous tumors cell populations (presented in Figs 1, 2 and 4) are available on GitHub at the following address: https://github.com/fallahi-sichani-lab/probabilisticDrugResponse.

## Supporting information

**S1 Fig. Probabilistic rate constants capture time-dependent heterogeneities in phenotypic responses. (A,B)** Simulation results showing changes in resistant enrichment ratio calculated for each of the $f_a$ metrics and for phenotype rate constants ($k_{death}$ and $k_{stasis}$) as a function of ω at a fixed value of $r = 16$ across different times of treatment. Data are shown for fixed inherent growth rates for the sensitive population, $k^S_{division\ (no\ drug)} = 0.035\ h^{-1}$ and two different rates of inherent growth for the resistant subpopulation: $k^R_{division\ (no\ drug)} = 0.009\ h^{-1}$ (A) and $k^R_{division\ (no\ drug)} = 0.035\ h^{-1}$ (B). **(C,D)** Simulation results showing changes in resistant enrichment ratio calculated for each of the $f_a$ metrics and for phenotype rate constants ($k_{death}$ and $k_{stasis}$) as a function of $r$ at a fixed value of ω = 0.03 across different times of treatment. Data are shown for fixed inherent growth rates for the sensitive population, $k^S_{division\ (no\ drug)} = 0.035\ h^{-1}$ and two different rates of inherent growth rate for the resistant subpopulation: $k^R_{division\ (no\ drug)} = 0.009\ h^{-1}$ (C) and $k^R_{division\ (no\ drug)} = 0.035\ h^{-1}$ (D). All data represent mean values from 50 simulations.
(TIF)

**S2 Fig. Sensitivity of $f_a$ metrics decreases as drug cytotoxicity increases. (A)** Input dose response profiles used in simulations. The maximum cytotoxic efficacy was varied at three different levels, whereas the cytostatic dose response profiles for all three conditions were held constant. **(B)** Model output measured from the simulated conditions in (A) at $t = 72$ h showing variations in viability, GR and the probabilistic phenotype rate constants. **(C)** Analysis of metric sensitivity with varying drug cytotoxicity parameter $P_{death}$, quantified per unit of time (h). Sensitivity analysis was performed on simulations with $P_{stasis} = 0$ and $k_{division\ (no\ drug)} = 0.035\ h^{-1}$. Initial cell number was $N_{live}(t = 0) = 5000$. Data shown are mean ± SEM across 50 simulations. Probabilistic phenotype rate constants were estimated from a 24 h time-interval centered at 72 h.
(TIF)

**S3 Fig. Overview of the time-lapse image analysis pipeline to quantify occurrence of single-cell phenotypic events from time-lapse live cell microscopy data.** The automated image analysis pipeline involves four steps: (1) Each background (BG) subtracted H2B image was segmented in CellProfiler for nucleus identification. For each nucleus object, a variety of features (e.g. mean signal intensities across multiple channels, area and shape) were measured. (2) To classify the phenotypes of interest (i.e. live or dead cells, Geminin^high or Geminin^low cells) in each image, classification models were trained in CellProfiler Analyst based on feature measurements of the user-annotated training sets. (3) Based on phenotype classifications of

individual cells for each image output from CellProfiler, corresponding synthetic images were generated in MATLAB for each phenotype of interest. Synthetic images contained synthetic pixels at locations of Geminin[high] or dead cells. To facilitate tracking of individual cells, relative intensities of the synthetic pixels for each phenotype were scaled with the mean intensity of the signal associated with that phenotype. For example, intensities of death synthetic pixels were scaled with the mean H2B signal intensities of individual cells, whereas intensities of the Geminin[high] synthetic pixels were scaled with the mean Geminin signal intensities. (4) Synthetic pixels for each phenotype were tracked separately in TrackMate. Since Geminin reporter level drops at the M phase, a division event is marked when the Geminin track ends. The beginning of a death track is also marked as a death event.
(TIF)

**S4 Fig. Probabilistic rate constants of phenotypic events measured using automated tracking is consistent with the rate constants acquired from manual single-cell tracking across different cell lines and drug conditions. (A-B)** Probabilistic rate constants of death ($k_{death}$) and division events ($k_{division}$) measured in (A) COLO858 and (B) MMACSF cells treated with Vemurafenib at the indicated doses, using automated tracking analysis pipeline (top row) versus manual tracking (bottom row) on the same set of time-lapse images. For each condition, the automated tracking estimates at each timepoint are the mean values across four replicated wells. Error bars represent SEM. The rate constants calculated from manual tracking data are based on individually tracked cells pooled from four replicated wells, including about 150–220 cells per condition.
(TIF)

**S5 Fig. Dynamic responses of COLO858 cells to epigenetic-modifying compounds and cell cycle inhibitors in sequential combination with Vemurafenib plus Trametinib.** Estimated dynamics of changes in live cell count, $k_{death}$, $k_{stasis}$ and $k_{division}$ measured from time-lapse live cell microscopy data for COLO858 cell responses to the combination of Vemurafenib (0.32 µM) and Trametinib (0.032 µM), a 3[rd] compound (including epigenetic-modifying compounds or cell cycle inhibitors), their triple combination, or vehicle (DMSO) control. Cells were treated initially for 24 h with DMSO control or one of the epigenetic-modifying compounds or cell cycle inhibitors (3[rd] compound) at the following concentrations: Fimepinostat (0.02 µM), Givinostat (0.2 µM), Birabresib (0.5 µM), I-BET762 (1 µM), SP2509 (1 µM), ORY-1001 (1 µM), JIB-04 (0.2 µM), CPI-455 (5 µM), AZ6102 (1 µM), NVP-TNKS656 (1 µM), Palbociclib (1 µM), and Abemaciclib (1 µM). After 24 h, Vemurafenib at 0.3 µM plus Trametinib at 0.03 µM, or DMSO control were added to each treatment condition. $k_{division\ (no\ drug)}$ used for the estimation of $k_{stasis}$ is estimated using cell division data averaged for the first 24 h in cells treated with DMSO only. In conditions where confluency was achieved, data-points were replaced with the last available data-point (dotted line). Data-points represent mean ± SEM across 2 or 3 replicates.
(TIF)

**S6 Fig. Dynamic responses of MMACSF cells to epigenetic-modifying compounds and cell cycle inhibitors in sequential combination with Vemurafenib plus Trametinib.** Estimated dynamics of changes in live cell count, $k_{death}$, $k_{stasis}$ and $k_{division}$ measured from time-lapse live cell microscopy data for MMACSF cell responses to the combination of Vemurafenib (0.32 µM) and Trametinib (0.032 µM), a 3[rd] compound (including epigenetic-modifying compounds or cell cycle inhibitors), their triple combination, or vehicle (DMSO) control. Cells were treated initially for 24 h with DMSO control or one of the epigenetic-modifying compounds or cell cycle inhibitors (3[rd] compound) at the following concentrations: Fimepinostat

(0.02 μM), Givinostat (0.2 μM), Birabresib (0.5 μM), I-BET762 (1 μM), SP2509 (1 μM), ORY-1001 (1 μM), JIB-04 (0.2 μM), CPI-455 (5 μM), AZ6102 (1 μM), NVP-TNKS656 (1 μM), Palbociclib (1 μM), and Abemaciclib (1 μM). After 24 h, Vemurafenib at 0.3 μM plus Trametinib at 0.03 μM, or DMSO control were added to each treatment condition. $k_{division\ (no\ drug)}$ used for the estimation of $k_{stasis}$ is estimated using cell division data averaged for the first 24 h in cells treated with DMSO only. In conditions where confluency was achieved, data-points were replaced with the last available data-point (dotted line). Data-points represent mean ± SEM across 2 or 3 replicates.

(TIF)

**S1 Table. Numerical data representing dynamic changes in live cell count and estimates of $k_{death}$ and $k_{division}$ measured from time-lapse live cell microscopy experiments for COLO858 cell responses to different doses of Vemurafenib.** Presented data are associated with Fig 3A.

(XLSX)

**S2 Table. Numerical data representing dynamic changes in live cell count and estimates of $k_{death}$ and $k_{division}$ measured from time-lapse live cell microscopy experiments for MMACSF cell responses to different doses of Vemurafenib.** Presented data are associated with Fig 3B.

(XLSX)

**S3 Table. Numerical data representing dynamic changes in live cell count and estimates of $k_{death}$ and $k_{division}$ measured from time-lapse live cell microscopy experiments for COLO858 cell responses to different drug combination treatment conditions.** Presented data are associated with Supporting Information S5 Fig.

(XLSX)

**S4 Table. Numerical data representing dynamic changes in live cell count and estimates of $k_{death}$ and $k_{division}$ measured from time-lapse live cell microscopy experiments for MMACSF cell responses to different drug combination treatment conditions.** Presented data are associated with Supporting Information S6 Fig.

(XLSX)

## Acknowledgments

We thank members of the Fallahi-Sichani laboratory, W Thomas, J Reyes, MW Covert, A Miyawaki, J Goedhart, A Bradley, R Benezra, V Becker, and Lahav and Sorger Laboratories at Harvard Medical School for help, discussion, software tools and reagents.

## Author Contributions

**Conceptualization:** Natacha Comandante-Lou, Mohammad Fallahi-Sichani.

**Formal analysis:** Natacha Comandante-Lou, Mohammad Fallahi-Sichani.

**Funding acquisition:** Mohammad Fallahi-Sichani.

**Investigation:** Natacha Comandante-Lou, Mehwish Khaliq, Divya Venkat, Mohan Manikkam, Mohammad Fallahi-Sichani.

**Methodology:** Natacha Comandante-Lou, Mehwish Khaliq, Divya Venkat, Mohan Manikkam, Mohammad Fallahi-Sichani.

**Software:** Natacha Comandante-Lou.

**Supervision:** Mohammad Fallahi-Sichani.

**Visualization:** Natacha Comandante-Lou, Mohammad Fallahi-Sichani.

**Writing – original draft:** Natacha Comandante-Lou, Mohammad Fallahi-Sichani.

**Writing – review & editing:** Natacha Comandante-Lou, Mehwish Khaliq, Divya Venkat, Mohan Manikkam, Mohammad Fallahi-Sichani.

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
