## [Decision Letter · Decision Letter 0]

11 Nov 2019

Dear Dr Fallahi-Sichani,

Thank you very much for submitting your manuscript, 'Phenotype-Based Probabilistic Analysis of Heterogeneous Responses to Cancer Drugs and Their Combination Efficacy', to PLOS Computational Biology. As with all papers submitted to the journal, yours was fully evaluated by the PLOS Computational Biology editorial team, and in this case, by independent peer reviewers. The reviewers appreciated the attention to an important topic but identified some aspects of the manuscript that should be improved.

We would therefore like to ask you to modify the manuscript according to the review recommendations before we can consider your manuscript for acceptance. Your revisions should address the specific points made by each reviewer and we encourage you to respond to particular issues Please note while forming your response, if your article is accepted, you may have the opportunity to make the peer review history publicly available. The record will include editor decision letters (with reviews) and your responses to reviewer comments. If eligible, we will contact you to opt in or out.raised.

- Supporting Information uploaded as separate files, titled 'Dataset', 'Figure', 'Table', 'Text', 'Protocol', 'Audio', or 'Video'.

We hope to receive your revised manuscript within the next 30 days. If you anticipate any delay in its return, we ask that you let us know the expected resubmission date by email at ploscompbiol@plos.org.

Sincerely,

Lingchong You

Associate Editor

PLOS Computational Biology

Jason Haugh

Deputy Editor

PLOS Computational Biology

[LINK]

Reviewer's Responses to Questions

**Comments to the Authors:**

Reviewer #1: In this study the authors develop a probabilistic description of drug-induced phenotypic events (cytotoxic and cytostatic events). They demonstrate, using a simulation of cell response to drug treatment, that probabilistic rate constants (kdeath, kstasis) are more sensitive to small, drug resistant cell populations. They also show that commonly used metrics (fraction of cells effected, fa) can bias the estimate of drug efficacy. Throughout the article, the authors use cancer drug resistance and the identification of drug combination therapies as the motivation for their study. The authors use time-lapse microscopy to measure the drug response of two cancer lines (both BRAF-mutated melanoma lines) and determine probabilistic rate constants from these movies. To determine the utility of their probabilistic phenotype metrics in evaluating drug combination efficacies, the authors develop a Bliss combination index (based on the widely referenced and used Bliss model of independence) that uses the probabilistic metrics to identify synergistic, independent, or antagonistic drug interactions. They use this model to evaluate the interaction of a Vemurafenib (BRAF inhibitor) and Trametinib (MEK inhibitor) treatment with 12 epigenetic-modifying and cell cycle inhibitors. These experiments lead to interesting observations of the differential effects of combination treatments on cytotoxicity vs cytostasis as well as the effects of drug treatment over time.

Overall, understanding how to evaluate the efficacy of drugs and drug interactions is an important and timely problem. In this paper, the authors focus on cancer drug resistance, but the evaluation of drugs and drug combinations is broadly important. Comparison of their metrics with the commonly used fa¬ and Bliss independence is well-motivated and should make this paper of interest to many researchers. I believe that after appropriate revision, this article could be appropriate for PLOS Computational Biology.

Strengths of the paper include, but are not limited to:

• The paper is well written. Supplemental Material and the Materials and Methods provide enough detail (except minor complaints as indicated in comments below).

• The model framework allows input parameters kdeath and kstasis to be chosen to represent drugs with a wide range of cytotoxic and cytostatic effects, and from these rate constants determine Pdeath and Pstasis. This is generally a useful framework.

• The authors demonstrate that the use of kstasis and k¬death in the Bliss combination index allow a variety of possible drug interactions to be distinguished, particularly those where drug combinations have uneven cytotoxic and cytostatic interactions.

• Twelve compounds chosen from epigenetic-modifying compounds and cell cycle inhibitors were assessed in combination with Vemurafenib (MEK inhibitor) and Trametinib (BRAF inhibitor). In addition to highlighting the utility of the technique, the observations regarding the CDK4/6 and BET inhibitors provide a hint of something interesting for future studies.

The following comments/questions should be addressed/answered to improve the manuscript:

• The derivation in “Stochastic simulation of cytotoxic and cytostatic drug effects” section assumes that the joint distribution of Pdivision(no drug) Pdivision(with drug) is the same as Pdivision(with drug), that is, a cell would never be more likely to divide with drug than without. Is this true? Are there no treatments that accelerate or force cells through a division? And would this matter? Please clarify.

• Fixed growth rates for the sensitive and resistant populations were picked to create Figure 2, with the resistant population assumed to grow approximately half as fast as the sensitive population. Can you clarify, does this affect the fa values and how “bad” they appear in terms of resistance enrichment ratio relative to the kstasis and kdeath metrics? It seems like it would matter since the fa metrics depend on time from treatment as well as population growth rate.

• Though it is clear from Figure 2 that the resistance enrichment ratio is smaller for the fa metrics, it isn’t clear to me that that means they aren’t distinguishable. A lower enrichment ratio doesn’t necessary mean that you can’t tell the different between samples, correct? That depends on the (experimental) variability? Perhaps I am missing something here.

• Does Figure 1D necessarily make the point that fa can lead to spurious results? The result we care about is whether or not drug interactions are detected. So the fact that fa metrics don’t necessarily match Pdeath and Pstasis doesn’t necessarily mean that the drug interaction results will be incorrect? We don’t expect fa to be the same as probabilities?

• The applicability of estimating phenotypic metrics from fluorescently labelled cell lines in time-lapse movies is not clear. The final paragraph of the discussion should be expanded to provide more compelling reasons and applications for this technique.

Minor comments:

• It might be clearer to label Pdivision as Pdivision(with drug) to make clear that this is the probability of cell division in the presence of drug. This is also what is done in the derivation in the Materials and Methods; it would be nice to have the variables be the same everywhere for consistency.

• You say “could be described” on p7 referring to the derivation of Pstasis. Is this because there are other reasonable ways to derive the equation, or because you are making a set of assumptions that are reasonable but might be missing something? Please clarify “could”, and the assumptions that you are making for this derivation either in the main text or in the derivation in the Materials and Methods. This also occurs in the derivation of the Bliss combination index (Eqn 12 and Eqn 13).

• The DIP and GR acronyms appear in the main text (p7) without being written out explicitly. I realize that they are in the Materials and Methods (p23) but it would be nice if the reader didn’t have to flip to the Methods to identify the acronym. Similarly, it would be nice for the reader if the equations for DIP and GR were repeated in the Materials and Methods (i.e. steady-state growth rate or first equation in ref. 5) so that we don’t have to go hunt them down in Harris, et al and Hafner, et al. It would also make the necessity of Eqn 17, 18 easier to understand.

• Figure 1d is difficult to understand. I am not completely sure what the easiest way to clarify the figure is? Maybe connecting lines of equal Pdeath (for example in the first figure) instead of having separate disconnected dots?

• The role of synthetic pixels in the image analysis and tracking program is not completely clear. I read it as synthetic pixels are used to mark dead cells or Geminin-High cells at a single timepoint (or timepoints where those classifications are true). But the how is this connected to the appropriate track of that cell before it died or went Geminin-High?

• In Equation 7, kdivision is ambiguous. Is this kdivision(with drug)?

• Is there a reason the growth timecourse for the control population is so much shorter (72 vs 120 hrs) than the drug treated samples?

• Line on p9 “In particular, as the efficiency of drug-induced cell killing increases, the sensitivity of fa metrics to detect drug tolerance in the surviving fraction of cells decreases”—did you justify this remark?

• A very minor point—the comma after your equations keeps moving to the beginning of the next sentence, which is slightly distracting.

Reviewer #2: This manuscript describes a novel mathematical framework for analyzing drug potency in a probabilistic fashion. Using simulations of stochastic cell death versus arrest of cell division, the authors inspect the typical multi-day clonal cytotoxicity assay used for evaluation of chemotherapeutics alone or in combination. Because arrest of cell division or cell death within a heterogeneous population can allow other tolerant cells to grow and replace within a colony, current metrics of evaluating drug responses are flawed.

I found the approach outlined in this well-written paper to be clever and an advance over the Bliss score currently in use for calculating drug interactions. Using live cell reporters and single cell tracking analysis, the authors estimate probabilistic rate constants and estimate the fraction of cells affected by drugs as a function of cell heterogeneity.

The modeling approach works amazingly well. While the methods detail the validation of a non-stationary Poisson process as the underlying assumption for their framework, I still wonder why this is applicable here. Our knowledge of the underlying mechanisms of apoptosis would suggest that the degree of cytotoxicity would initiate release of pro-apoptotic factors that would influence cell death in a manner that isn’t just time-dependent. A more detailed discussion of this would be helpful for the reader to reconcile the appropriateness of the model assumption, especially for different initial seeding densities of cells.

Minor revision, Fig 5A: Hierarchical clustering should be represented with a dendrogram on heat map

**Have all data underlying the figures and results presented in the manuscript been provided?**

Reviewer #1: No: Model code and files as well as data extracted from imaging timecourses was not available with my review. These should be made available if the article is published.

Reviewer #2: Yes

PLOS authors have the option to publish the peer review history of their article (what does this mean?). If published, this will include your full peer review and any attached files.

Reviewer #1: No

Reviewer #2: No

---

## [Decision Letter · Decision Letter 1]

27 Jan 2020

Dear Dr. Fallahi-Sichani,

We are pleased to inform you that your manuscript 'Phenotype-Based Probabilistic Analysis of Heterogeneous Responses to Cancer Drugs and Their Combination Efficacy' has been provisionally accepted for publication in PLOS Computational Biology.

Before your manuscript can be formally accepted you will need to complete some formatting changes, which you will receive in a follow up email. A member of our team will be in touch within two working days with a set of requests.

Best regards,

Lingchong You

Associate Editor

PLOS Computational Biology

Jason Haugh

Deputy Editor

PLOS Computational Biology

Reviewer's Responses to Questions

**Comments to the Authors:**

Reviewer #1: In combination therapies single-cell and subpopulation effects can diminish therapeutic efficiency. Current screening of drug combinations obscures these effects. In this article, the authors propose probabilistic metrics that deconvolve drug effect on killing versus effects on growth and show that using these metrics increases the sensitivity of short-term response assays to detect cell to cell heterogeneity and drug resistant populations.

In this revision, the authors have addressed the major and minor concerns I expressed in my first review as follows:

- They have clarified that Pstasis=0 characterizes both drugs that do not inhibit cell division as well as drugs that theoretically could accelerate cell division.

- Clarified the use of synthetic pixels in their image analysis through additional text in the Methods section

- Defined DIP and GR within the Methods section

- Added an additional supplemental figure (S1) to clarify how the growth rate of the resistant population affects the sensitivity of their probabilistic metrics.

- Defined fractional sensitivity in the methods to compare the sensitivity of different metrics, rather than the “distinguishability”, which is not illustrated in Supplemental Figure 2.

- Clarified the caption to Figure 2

- Clarified Figure 1D with dashed lines

In my opinion, this article is appropriate for publication in PLOS Computational Biology.

Reviewer #2: The authors did an excellent job addressing critiques.

**Have all data underlying the figures and results presented in the manuscript been provided?**

Reviewer #1: Yes

Reviewer #2: Yes

PLOS authors have the option to publish the peer review history of their article (what does this mean?). If published, this will include your full peer review and any attached files.

Reviewer #1: No

Reviewer #2: No

---

## [Editor Report · Acceptance letter]

14 Feb 2020

PCOMPBIOL-D-19-01684R1 

Phenotype-Based Probabilistic Analysis of Heterogeneous Responses to Cancer Drugs and Their Combination Efficacy

Dear Dr Fallahi-Sichani,

I am pleased to inform you that your manuscript has been formally accepted for publication in PLOS Computational Biology. Your manuscript is now with our production department and you will be notified of the publication date in due course.

With kind regards,

Laura Mallard
